# Pharmacological Cardioversion in Patients with Recent-Onset Atrial Fibrillation and Chronic Kidney Disease Subanalysis of the CANT II Study

**DOI:** 10.3390/ijerph19084880

**Published:** 2022-04-17

**Authors:** Beata Ceynowa-Sielawko, Maciej T. Wybraniec, Aleksandra Topp-Zielińska, Aleksander Maciąg, Dawid Miśkowiec, Paweł Balsam, Maciej Wójcik, Wojciech Wróbel, Michał M. Farkowski, Edyta Ćwiek-Rębowska, Krzysztof Ozierański, Robert Błaszczyk, Karolina Bula, Tomasz Dembowski, Michał Peller, Bartosz Krzowski, Wojciech Wańha, Marek Koziński, Jarosław D. Kasprzak, Hanna Szwed, Katarzyna Mizia-Stec, Marek Szołkiewicz

**Affiliations:** 1Department of Cardiology and Interventional Angiology, Kashubian Center for Heart and Vascular Diseases, Pomeranian Hospitals, 84-200 Wejherowo, Poland; beataceynowa@interia.pl (B.C.-S.); oleanderka@wp.pl (A.T.-Z.); 2First Department of Cardiology, School of Medicine in Katowice, Medical University of Silesia, Upper Silesia Medical Centre, 40-635 Katowice, Poland; maciejwybraniec@gmail.com (M.T.W.); wojtekwrobel@poczta.onet.pl (W.W.); karolina.bula@yahoo.pl (K.B.); kmiziastec@gmail.com (K.M.-S.); 3Club 30′ of the Polish Cardiac Society, 00-193 Warsaw, Poland; dawid.miskowiec@gmail.com (D.M.); pawel@balsam.com.pl (P.B.); m.wojcik@umlub.pl (M.W.); mfarkowski@gmail.com (M.M.F.); krzysztof.ozieranski@gmail.com (K.O.); robertblaszczyk1@wp.pl (R.B.); wojciech.wanha@gmail.com (W.W.); marekkozinski@wp.pl (M.K.); kasprzak@ptkardio.pl (J.D.K.); 42nd Department of Heart Arrhythmia, National Institute of Cardiology, 04-628 Warsaw, Poland; maciag_o@poczta.onet.pl; 5Department of Cardiology, Medical University of Lodz, 91-347 Lodz, Poland; edyta.cwiek.rebowska@gmail.com (E.Ć.-R.); tomaszdembowski@gmail.com (T.D.); 61st Chair and Department of Cardiology, Medical University of Warsaw, 02-097 Warsaw, Poland; michalpeller@gmail.com (M.P.); bartekkrzowski@gmail.com (B.K.); 7Chair and Department of Cardiology, Medical University of Lublin, 20-090 Lublin, Poland; 8Division of Cardiology and Structural Heart Diseases, Medical University of Silesia, Upper Silesia Medical Center, 40-635 Katowice, Poland; 9Department of Cardiology and Internal Medicine, Medical University of Gdansk, 81-519 Gdynia, Poland; 10Department of Coronary Artery Disease and Cardiac Rehabilitation, National Institute of Cardiology, 04-628 Warsaw, Poland; hszwed@ikard.waw.pl

**Keywords:** atrial fibrillation, pharmacological cardioversion, chronic kidney disease, antazoline, propafenone, amiodarone

## Abstract

Pharmacological cardioversion (PCV) is commonly a primary option for termination of recent-onset atrial fibrillation (AF) in emergency departments (ED). This is a subanalysis of the CANT II study, evaluating the effectiveness and safety of antazoline in patients (*n* = 777) at three stages of chronic kidney disease (CKD): Group I > 60 mL/min (*n* = 531), Group II 45–59 mL/min (*n* = 149), and Group III < 45 mL/min (*n* = 97). Patients in Group III were older and with a higher prevalence of co-morbidities; however, we did not find statistically significant differences in the overall effectiveness of PCV in comparison with the other groups. In patients receiving amiodarone, the PCV success rate was similar in all the studied groups, but along with a renal function decline, it decreased in patients receiving antazoline (79.1 vs. 35%; *p* < 0.001), and it increased almost significantly in patients receiving propafenone (69.9 vs. 100%; *p* = 0.067). In patients in Group I, antazoline restored a sinus rhythm as effectively as propafenone and amiodarone; however, in patients in Group III, both antazoline and amiodarone became less effective in restoring a sinus rhythm than propafenone (*p* = 0.002 and *p* = 0.034, respectively). The rate of safety endpoint was the highest in patients in Group III (eGFR < 45 mL/min), and it was significantly higher than in patients in Groups I and II (*p* = 0.008 and *p* = 0.036, respectively). We did not observe antazoline-related adverse events in any of the studied groups of patients. This real-world registry analysis revealed a different influence of CKD on the effectiveness of individual drugs, and while propafenone and amiodarone maintained their AF termination efficacy, antazoline became significantly less effective in restoring sinus rhythm.

## 1. Introduction

Atrial fibrillation (AF) is the most common supraventricular arrhythmia. The rate of its spreading is enormous; thus, it has become a crucial problem for all healthcare systems and a socio-economic burden all over the world [1]. It is influential enough to reduce the prognostic power of cardiovascular event predictors, but, above all, it significantly increases the risk of death [2,3]. The management of AF with the rate control strategy offers no survival advantage over a rhythm control one, but the latter is still preferred by most patients [4]. This is one of the reasons why the number of admissions to emergency departments for AF termination is exceptionally high.

Electrical cardioversion (ECV) and pharmacological cardioversion (PCV) are common methods used world-wide to terminate atrial fibrillation and to restore a physiological sinus rhythm. ECV requires short-term general anaesthesia and should be preceded by at least 6 h fasting; thus, a pharmacological approach is usually the primary option in most cases. Anti-arhythmic drugs (AADs) variously modify the ion channel function and/or intracellular mechanisms regulated by adrenergic activity. They are classified according to the Vaughan–Williams classification. Administration of class IA AAD (e.g., quinidine and disopyramide) or class IC AAD (e.g., propafenone and flecainide), with a dominant sodium channel-blocking activity, is often successful, although pro-arrhythmia is a noted phenomenon, particularly in heart failure patients [5]. Amiodarone—a class III AAD affecting a broad range of channels, including potassium, sodium, and calcium channels, as well as blocking α- and β-receptors—has a delayed onset of action [6]. The new agent vernakalant, with an atrial-selective potassium channel-blocking activity, does not actually fit in the Vaughan–Williams classification. It is perceived as highly effective, but it is not widely available [7]. Antazoline seems to be a reasonable alternative. Its antiarrhythmic quinidine-like and anticholinergic properties were discovered in the middle of the last century; however, the lack of randomised trials confirming its efficacy and safety has prevented the drug from being listed in official guidelines. Nevertheless, it is registered and has been widely used to terminate supraventricular arrhythmias in some countries, since observational studies showed its almost immediate onset of action, namely, an efficacy up to 80% with successful cardioversion within several minutes, and excellent tolerance and safety [8]. Maciag et al. reported the results of the first randomised study proving that intravenous antazoline is safe and significantly superior to placebos for the termination of recent-onset AF [9], while Farkowski et al. demonstrated that antazoline is more effective in restoring sinus rhythm than propafenone [10]. Recently, we presented probably the largest analysis of a multicentre, retrospective, real-world registry for pharmacological cardioversion (PCV) in patients with recent-onset AF (CANT II Study). Intravenous antazoline had the highest PCV success rate of the studied drugs and was superior to amiodarone and comparable to propafenone [11].

Patients with chronic kidney disease (CKD) are a highly specific population, with structural and functional disorders in all organs and tissues, and CKD itself is one of the well-defined factors promoting the development and recurrence of AF. Statistical data show that AF can be found in 20% of CKD patients [12]. CKD is characterised by a high morbidity and overall mortality. Its prevalence increases along with the ageing of a population. All CKD-related natural pathologies, all biological and haemodynamic consequences of the disease (including the enhanced activity of RAA system in particular), a permanent inflammatory state, increased tissue and organ fibrosis, excessively accumulated amyloid, and induced cell necrosis and apoptosis strongly contribute to a multi-level structural, functional, metabolic, neurohormonal and electric atrial remodelling, and this results in a high prevalence of AF recurrence and a high rate of its progression to a permanent type [13]. The efficacy of many drugs, including antiarrhythmic ones, has not been extensively studied in CKD, and, thus, one may question if they present the same degree of efficiency and safety in patients with this disease.

In clinical trials, CKD is usually analysed as one of many variables impacting the final results. There are no published reports relating to the efficiency of PCV in patients with paroxysmal AF (lasting up to 7 days) at different stages of CKD. Thus, the objective of this study was to evaluate the safety and effectiveness of PCV (primary aim), in particular with intravenous antazoline in comparison to amiodarone, propafenone, and/or overlapping therapy (secondary aim), administered to patients with CKD admitted to the emergency department for termination of AF.

## 2. Materials and Methods

We analysed the pooled data of the Cardioversion with an intravenous ANTazoline mesylate (CANT II) study, which evaluated the safety and effectiveness of intravenous antazoline in patients with recent-onset AF based on the multicentre, retrospective, real-world registry [11]. This registry included the data of patients (*n* = 1365) with paroxysmal/persistent AF, admitted to the emergency department for an urgent restoration of sinus rhythm. The present analysis covered *n* = 777 patients with available data on serum creatinine concentration and estimated glomerular filtration rate (eGFR).

The study protocol was accepted by the local ethics committee, and all the patients participating in the registry provided informed written consent. The fundamental inclusion criterion was a pharmacologically approached recent-onset AF to restore a sinus rhythm. The exclusion criteria were permanent AF, paroxysmal arrhythmia lasting more than 48 h, not being effectively covered by chronic anticoagulation, symptomatic arrhythmia urgently requiring electrical cardioversion or pacemaker implantation, chronic antiarrhythmic therapy (except β-blockers), and any other condition that in the opinion of the attending physician might have negatively affected the patients’ safety.

The description of data acquisition and the definitions used have previously been published [11]. We used CKD-EPI (chronic kidney disease epidemiology collaboration) equations to estimate eGFR expressed in mL/min/1.73 m^2^ and, thus, the stage of the disease. We analysed and compared the results concerning the safety and effectiveness of intravenous antazoline in populations of patients at 3 stages of CKD defined on a basis of eGFR: >60 mL/min (*n* = 531), 45–59 mL/min (*n* = 149), and <45 mL/min (*n* = 97). An insufficient number of patients prevented us from creating a separate group of those with extremely advanced CKD (<30 mL/min), and, therefore, all these patients were included in Group III. Otherwise, it might have affected the reliability of the results.

The primary endpoint was the termination of AF and the restoration of a sinus rhythm persisting until discharge. PCV was found to be ineffective if a sinus rhythm had not been restored during a 12 h period or if an electrical cardioversion had been performed. The safety endpoint was any adverse event reported after starting PCV with antiarrhythmic drugs, including bradycardia < 45 bpm, hypotension (SBP drop > 40 mmHg), syncope, and death.

The allocation of antiarrhythmic drugs, their dose, and concomitant medications were left to the discretion of the attending physician; however, the latest European Society of Cardiology (ESC) guidelines for the diagnosis and management of AF had to be taken into account. According to the registry data, the patients received intravenous antazoline mesylate (Phenazolinum, Polfa, Warsaw, Poland), intravenous amiodarone (Cordarone, Sanofi-Aventis France, Gentilly, France), intravenous/oral propafenone hydrochloride (Rytmonorm, Mylan Healthcare, Warsaw, Poland), or a combination of these agents (overlapping therapy). Antazoline was administered as a single or repeated intravenous bolus of 100–200 mg, commonly diluted in 100 mL of 0.9% NaCl. Amiodarone was administered in a 5% glucose solution, usually preceded by an intravenous bolus of 150 mg. Propafenone was given orally in 150 mg pills or as a slow intravenous bolus of 70 mg, diluted in 100 mL of 0.9% NaCl.

Statistical analysis was performed using SPSS v.25.0 software (IBM Corp., Armonk, NY, USA) and MedCalc v.14.8.1 software (MedCalc Software, Ostend, Belgium). In the case of normal distribution, Student’s t-tests or analysis of variance (ANOVA) tests were applied, while in non-normally distributed variables, two-tailed Mann–Whitney U or Kruskal–Wallis tests were utilised. The significance of proportions in contingency tables was calculated using a chi-square test with Bonferroni adjustment. All the variables with *p* < 0.1 in univariate analysis were incorporated into a logistic regression model so as to establish independent predictors of successful PCV. *p* < 0.05 was regarded as a statistically significant level throughout the analyses.

## 3. Results

A total of *n* = 1365 patients were included in the CANT II study. Of these, *n* = 777 had a determined eGFR and they were enrolled into the present analysis. The patients were divided into three groups according to the stage of CKD: Group I—eGFR > 60 mL/min (*n* = 531), Group II—eGFR 45–59 mL/min (*n* = 149), and Group III—eGFR < 45 mL/min (*n* = 97). Patients with the most advanced CKD (Group III) were older than patients in Groups I and II (mean age: 74.04 ± 10.5 vs. 64.3 ± 12.2 and 71.0 ± 9.7 years, respectively), and they had a greater prevalence of arterial hypertension (91.7% vs. 74.8% and 83.8%), diabetes (34.4% vs. 20.6% and 20.9%), coronary and/or peripheral vascular disease (42.3% vs. 26.9% and 35.1%), and history of ischaemic stroke/TIA (18.6% vs. 6.4% and 6.8%), as well as a higher CHA_2_DS_2_-VASc score (4.4 ± 1.4 vs. 2.7 ± 1.6 and 3.6 ± 1.6, respectively).

We also found a greater percentage of women in Group III. This phenomenon is difficult to explain since epidemiological data indicate that male patients show a higher incidence of advanced CKD and atrial fibrillation. Presumably, this is specific to our cohort. The baseline characteristics of all study patients depending on their CKD stage are presented in Table 1.

The analysis of different predictors of successful pharmacological cardioversion is presented in Table 2. The univariate analysis revealed that eGFR, the administration of intravenous potassium, left atrial diameter, left ventricular ejection fraction, EHRA score, heart rate, serum potassium concentration, the presence of coronary artery disease, and AF episode duration predicted the successful termination of AF (Table 2). The logistic regression analysis indicated that shorter AF episode duration, greater heart rate, and LVEF independently predicted the return of sinus rhythm (AUC for the model: 0.709; 95% CI-0.65-0.76; Hosmer–Lemeshow test *p* = 0.0.378).

We did not find statistically significant differences in the overall effectiveness of PCV between the studied groups. We have observed a tendency towards a less effective PCV in patients with more advanced CKD; however, the acquired differences did not reach a threshold for statistical significance (Figure 1).

Instead, we found significant differences in terms of effectiveness of individual drugs. In patients receiving amiodarone, the success rate of cardioversion was similar in all the studied groups; however, along with a decline in renal function, it decreased in patients receiving antazoline (79.1 vs. 35%; *p* < 0.001), and it increased almost significantly in patients receiving propafenone (69.9 vs. 100%; *p* = 0.067). The results are presented in Figure 2.

Comparing the effectiveness of antazoline with the other studied drugs at different stages of CKD (Figure 3), we found that in patients in Group I (eGFR ≥ 60 mL/min), antazoline restored a sinus rhythm as effectively as propafenone and amiodarone. In fact, we noted a difference suggesting the superiority of antazoline over amiodarone, but at the borderline of statistical significance (*p* = 0.052). When the patients in Group III (eGFR < 45 mL/min) were considered, both antazoline and amiodarone became less effective in restoring a sinus rhythm than propafenone (*p* = 0.002 and *p* = 0.034, respectively).

We also compared the PCV success rate for antazoline and overlapping therapy (Figure 4). In patients in Group I (eGFR ≥ 60 mL/min), antazoline was significantly more effective in the termination of recent-onset AF than overlapping therapy (*p* < 0.001), but it became inferior (*p* < 0.03) in patients in Group III (eGFR < 45 mL/min).

The composite safety endpoint was reported in 17 patients (2.19%). The most frequent was bradycardia, which occurred in 12 out of 17 patients (70.6%). There was no report of hospital deaths. The rate of safety endpoint was the highest in patients in Group III (eGFR < 45 mL/min), and it was statistically significantly higher than in patients in Groups I and II (*p* = 0.008 and *p* = 0.036, respectively; Table 1). We did not observe antazoline-related adverse events in any of the studied patients with a determined eGFR.

## 4. Discussion

Our analysis showed that PCV used to terminate AF might be less effective in patients with advanced CKD. The noted difference did not quite reach the threshold for statistical significance, although it indicated a tendency towards significance. This is not surprising if one takes into account that in the setting of CKD, many various pathological processes accelerate, and a remodelling of tissues and organs is more pronounced, which impairs or sometimes even stops an effective treatment. For instance, it is worth noting a higher prevalence of left atrium low-voltage zones among patients with CKD [14]. These zones represent structural remodelling and are known as a strong predictor for AF recurrence [15]. Published reports prove that in patients with CKD, AF recurrence rates after pulmonary vein isolation performed either with cryoballoon or radiofrequency energy are higher, and the AF-free survival rate is shorter than in patients without the disease [16,17]. Thus, CKD appears not to be a desired environment for AF, since it promotes its recurrence rate and resistance to cardioversion, as well as accelerating its progression to a sustained/permanent form.

Antazoline proved to be a surprisingly efficacious and safe agent for the termination of recent-onset AF. Previously published reports demonstrated its impact on ECG parameters (prolongation of *p*-wave, PR and QRS duration, and QTc interval) without any significant influence on cardiac output, total peripheral resistance, or blood pressure [18]. Moreover, other clinical and experimental studies indicated that it led to an increase in atrial post-repolarisation refractoriness or an interatrial conduction time, and, thus, it reduced AF inducibility [19]. Admittedly, antazoline is currently not listed in official guidelines, but it is worth noting its high effectiveness in restoring a sinus rhythm. These properties of antazoline were also validated in the CANT II study, in which antazoline proved to be the most successful drug in terminating short-term AF and in restoring a sinus rhythm in the entire spectrum of the study-participating patients; it was better than amiodarone and comparable with propafenone [11]. However, in the present subanalysis of the CANT II study, we observed similar effectiveness of antazoline only in patients with eGFR > 60 mL/min. Its effectiveness significantly decreased along with a decline in renal function, and, eventually, in patients with more advanced kidney disease (eGFR < 45 mL/min), antazoline became significantly inferior to both propafenone (*p* = 0.002) and overlapping therapy (*p* = 0.025), while it was non-significantly inferior to amiodarone (*p* = 0.067). We do not know the precise reasons for the impaired antazoline efficiency found in patients with advanced CKD; however, this phenomenon cannot be overlooked. Apparently, it loses its unique abilities in a CKD environment, and this should be taken into consideration when making treatment decisions. It was not surprising that patients with advanced CKD were older and had more co-morbidities; all these elements are biologically and medically closely related to each other. Indeed, this could have had an impact on the treatment results. It is worth noting, however, that antazoline was the only drug with a worsening efficacy profile. Thus, one cannot exclude the likelihood that ageing and co-morbidities affect the efficiency of antazoline more than they do other AAD. The drug has not yet been well investigated; in some countries, it still has not been registered for arrhythmia termination. The overall number of published reports is limited, and there are none on patients with CKD. Napoli et al. indicated that propafenone might be truly effective in restoring a sinus rhythm in patients with AF and CKD, including patients on maintenance haemodialysis treatment (87%; 21/24 pts) [20]. The results of this small observational study are consistent with our analysis, which revealed that propafenone was highly effective in the termination of recent-onset AF in both the whole population of the CANT study and varied stages of CKD. Therefore, one may conclude that propafenone, and not antazoline, should be considered first if we decide to pharmacologically terminate recent-onset AF in patients with advanced CKD.

It needs to be emphasised that the overall number of reported adverse events was limited, and, therefore, the results concerning all antazoline and the other drugs’ safety profiles should be interpreted with caution. However, the highest rate of safety endpoints was observed in the patients with the most advanced CKD. Antazoline was entirely safe; we did not observe any antazoline-related adverse events in any of the studied groups of patients. These data confirm antazoline’s highly favourable safety profile, which has been reported previously [9,10].

## 5. Study Limitations

There are differences between the groups in terms of baseline cardiovascular risk. It is noticeable that the patients in Group III, with the most advanced CKD, are older and have a higher prevalence of co-morbidities, which may affect any treatment outcome. However, this is an analysis of retrospective real-world registry data, and, therefore, the population of patients included could not be truly homogenous.

In spite of the fact that this is the largest analysis of the effectiveness and safety of antazoline in the termination of recent-onset AF in patients with different stages of CKD, the study sample of patients with the most advanced CKD was highly limited and not sufficiently representative. Therefore, we were not able to provide reliable statistical data for this specific subpopulation.

## 6. Conclusions

In summary, we conclude that our real-world registry analysis demonstrated high effectiveness of PCV in patients with recent-onset AF; however, it seemed to decline along with the decrease in glomerular filtration rates. The detailed analysis revealed a different influence of CKD on individual drug effectiveness, and while propafenone and amiodarone preserved their well-known AF termination efficacy, antazoline seemed to be less effective among patients with eGFR < 45 mL/min/1.73 m^2^. Nevertheless, its excellent safety profile remained unchanged. Further studies are needed to confirm these findings.

## Figures and Tables

**Figure 1 ijerph-19-04880-f001:**
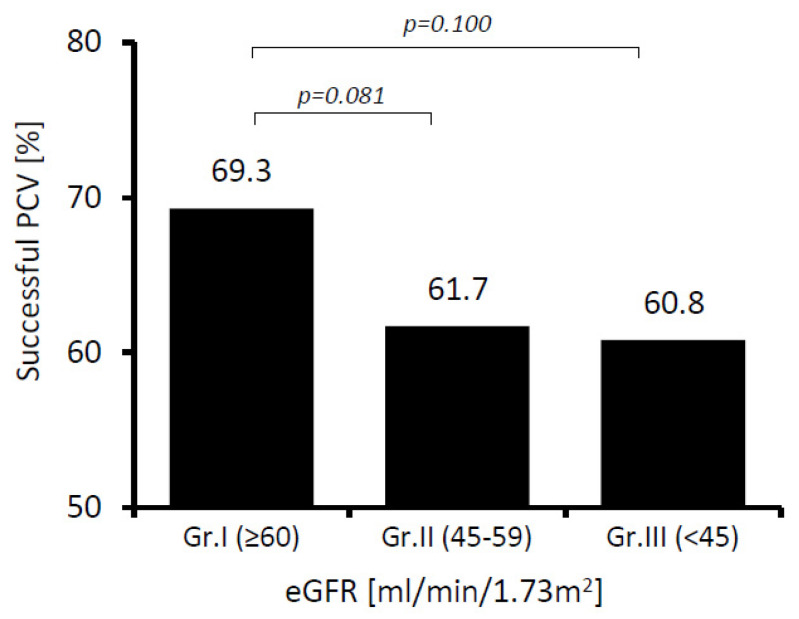
Effectiveness of pharmacological cardioversion (PCV) at different stages of chronic kidney disease. One may see a tendency towards worse PCV effectiveness along with a decrease in eGFR, but the differences between groups are not statistically significant (patients: Group I *n* = 531; Group II *n* = 149; Group III *n* = 97).

**Figure 2 ijerph-19-04880-f002:**
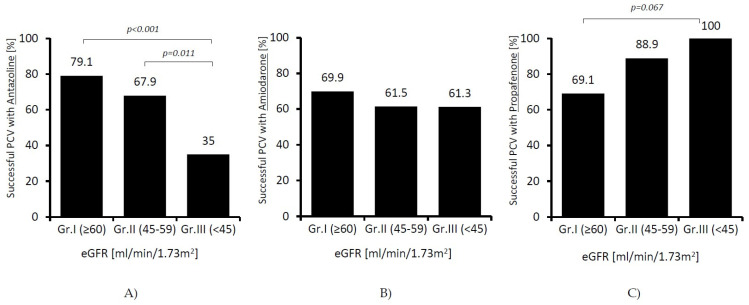
Effectiveness of pharmacological cardioversion (PCV) with antazoline (**A**), amiodarone (**B**), and propafenone (**C**) at different stages of chronic kidney disease (patients: Group I *n* = 531; Group II *n* = 149; Group III *n* = 97).

**Figure 3 ijerph-19-04880-f003:**
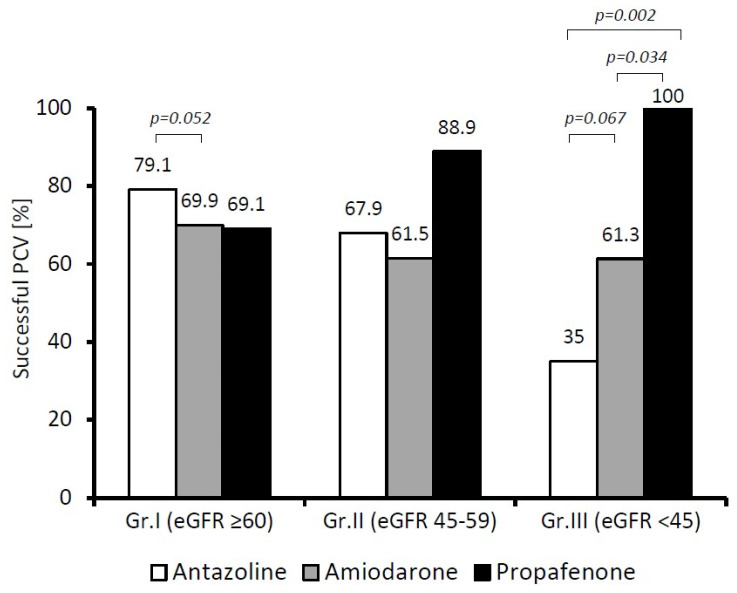
Comparison of pharmacological effectiveness (PCV) of the drugs at different stages of chronic kidney disease (patients: Group I *n* = 531; Group II *n* = 149; Group III *n* = 97).

**Figure 4 ijerph-19-04880-f004:**
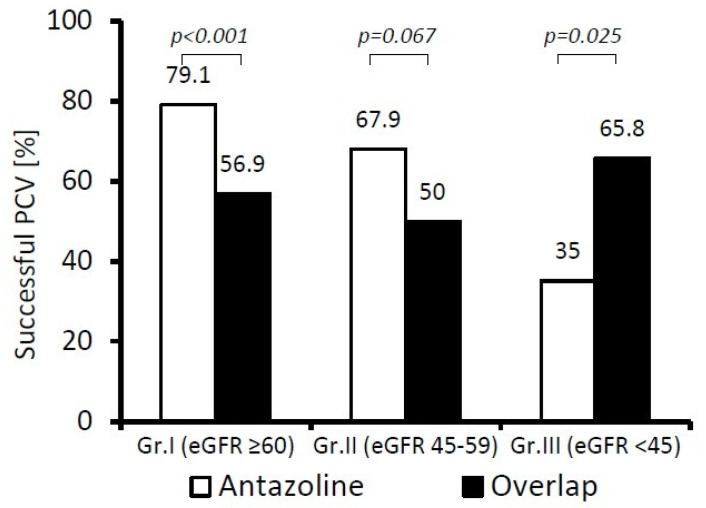
Comparison of pharmacological cardioversion (PCV) effectiveness of antazoline and overlapping therapy at different stages of chronic kidney disease (patients: Group I *n* = 531; Group II *n* = 149; Group III *n* = 97).

**Table 1 ijerph-19-04880-t001:** Clinical characteristics of studied patients depending on the stage of chronic kidney disease (CAD—coronary artery disease; PAD—peripheral artery disease; TIA—transient ischaemic attack; eGFR—estimated glomerular filtration rate; BMI—body mass index; CHA_2_DS_2_-VASc—scoring system estimating a risk of stroke in patients with AF; EHRA—European Heart Rhythm Association).

Variable	Group I eGFR ≥ 60 (*n* = 531)	Group II eGFR 45–59 (*n* = 149)	Group III eGFR < 45 (*n* = 97)	Statistical Analysis (*p*)			
	*n* (%)	*n* (%)	*n* (%)	I–III	I vs. II	I vs. III	II vs. III
Male sex	282 (53.7)	58 (39.5)	23 (23.7)	<0.001	<0.001	<0.001	0.013
Age > 65 (years)	278 (52.3)	123 (82.5)	80 (82.5)	<0.001	<0.001	<0.001	0.988
Age > 75 (years)	90 (16.9)	57 (38.3)	51 (52.6)	<0.001	<0.001	<0.001	0.027
Hypertension	397 (74.8)	124 (83.8)	89 (91.7)	<0.001	0.022	<0.001	0.070
Diabetes	109 (20.6)	31 (20.9)	33 (34.4)	0.010	0.920	0.003	0.020
CAD/PAD	143 (26.9)	52 (35.1)	41 (42.3)	0.004	0.051	0.002	0.261
History of stroke/TIA	34 (6.4)	10 (6.8)	18 (18.6)	<0.001	0.877	<0.001	0.005
Oral anticoagulants	357 (69.0)	109 (75.2)	77 (85.6)	0.004	0.154	<0.001	0.057
Beta-blocker use	184 (34.9)	57 (38.3)	30 (31.2)	0.526	0.452	0.487	0.263
Amiodarone	173 (32.6)	39 (26.2)	31 (32.0)	0.325	0.136	0.904	0.326
Propafenone	42 (7.9)	9 (6.0)	8 (8.2)	0.724	0.444	0.910	0.505
Antazoline	172 (32.4)	53 (35.6)	20 (20.6)	0.036	0.466	0.021	0.012
Overlap	144 (27.1)	48 (32.2)	38 (39.2)	0.042	0.222	0.016	0.263
Successful PCV	368 (69.3)	92 (61.7)	59 (60.8)	0.092	0.081	0.100	0.885
Safety endpoint	9 (1.7)	2 (1.34)	6 (6.2)	0.015	0.763	0.008	0.036
Death	0 (0.0)	0 (0.0)	0 (0.0)	-	-	-	-
Bradycardia < 45 bpm	6 (1.13)	1 (0.7)	4 (4.1)	0.050	0.624	0.030	0.061
Syncope	1 (0.19)	0 (0.0)	0 (0.0)	0.793	0.596	0.669	-
Hypotension	2 (0.4)	1 (0.7)	1 (1.0)	0.680	0.633	0.391	0.759
**Variable**	**Group I**	**Group II**	**Group III**	**Statistical Analysis (*p*)**			
	**Mean ± SD**	**Mean ± SD**	**Mean ± SD**	**I–III**	**I vs. II**	**I vs. III**	**II vs. III**
Age (years)	64.3 ± 12.2	71.0 ± 9.7	74.0 ± 10.5	<0.001	<0.001	<0.001	0.013
eGFR (mL/min)	80.7 ± 12.3	53.6 ± 4.2	36.9 ± 7.1	<0.001	<0.001	<0.001	<0.001
BMI	27.8 ± 4.4	28.7 ± 3.9	28.5 ± 5.3	0.477	0.212	0.776	0.814
CHA_2_DS_2_VASc	2.7 ± 1.6	3.6 ± 1.6	4.4 ± 1.4	<0.001	<0.001	<0.001	<0.001
EHRA score	2.5 ± 0.7	2.6 ± 0.8	2.2 ± 0.9	0.005	0.043	0.021	0.003
Drug (total dose):							
Amiodarone (mg)	509.0 ± 258.2	489.2 ± 257.6	422.4 ± 251.5	0.045	0.494	0.014	0.110
Propafenone (mg)	265.0 ± 210.9	288.1 ± 247.1	289.4 ± 227.6	0.937	0.726	0.909	0.823
Antazoline (mg)	214.3 ± 78.8	231.0 ± 81.3	209.4 ± 71.4	0.283	0.167	0.653	0.151

**Table 2 ijerph-19-04880-t002:** Parameters of logistic regression analysis: AUC—0.709; 95% CI—0.65-0.76; Hosmer–Lemeshow test *p* = 0.0.378 (BMI—body mass index; CAD—coronary artery disease; PAD—peripheral artery disease; eGFR—estimated glomerular filtration rate; CHA_2_DS_2_-VASc—scoring system estimating a risk of stroke in patients with AF; EHRA—European Heart Rhythm Association; LAd-PLAX—left atrium dimension–parasternal long-axis echocardiogram view; LVEF—left ventricular ejection fraction; PVI—pulmonary veins isolation; TIA—transient ischaemic attack).

Variable	Univariate Analysis			Logistic Regression Analysis		
	OR	95% CI	*p* Value	OR	95% CI	*p* Value
Male sex	0.94	0.74–1.19	0.598	-	-	-
Age (per 1 year)	0.99	0.98–1.00	0.286	-	-	-
BMI (per 1 kg/m^2^)	0.98	0.92–1.04	0.511	-	-	-
Diabetes	0.93	0.69–1.26	0.646	-	-	-
CAD/PAD	1.55	1.18–2.04	0.002	-	-	-
Hypertension	0.81	0.58–1.12	0.209	-	-	-
eGFR (per 1 mL/min)	1.01	1.00–1.02	0.023	-	-	-
CHA_2_DS_2_-VASc (per 1 point)	0.96	0.89–1.05	0.403	-	-	-
AF episode duration (per 1 h)	0.99	0.98–1.00	0.001	0.99	0.98–1.00	0.008
EHRA score	0.66	0.55–0.80	<0.001	-	-	-
WBC (per 1000/mm^3^)	0.99	0.94–1.05	0.862	-	-	-
Haemoglobin (per 1 g/dL)	0.92	0.84–1.01	0.075	-	-	-
Heart rate (per 1 bpm)	1.01	1.00–1.01	0.003	1.01	1.00–1.02	0.036
Potassium concentration (per 1 mEq/L)	0.66	0.49–0.88	0.005	-	-	-
LAd PLAX (per 1 mm)	0.94	0.91–0.97	<0.001	-	-	-
LVEF (per 1%)	1.03	1.02–1.05	<0.001	1.06	1.02–1.10	0.001
History of PVI	1.36	0.88–2.10	0.166	-	-	-
Structural heart disease	1.18	0.93–1.51	0.176	-	-	-
Troponin concentration (per 1 ng/mL)	1.29	0.34–4.86	0.709	-	-	-
History of stroke/TIA	1.67	0.91–3.07	0.099	-	-	-
Beta-blocker use	1.01	0.80–1.29	0.919	-	-	-

## Data Availability

The data supporting the findings of the study are available from Maciej T. Wybraniec (maciejwybraniec@gmail.com) on reasonable request.

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
