# Peer review of "Pharmacological Cardioversion in Patients with Recent-Onset Atrial Fibrillation and Chronic Kidney Disease Subanalysis of the CANT II Study"

_ijerph, 2022, doi:10.3390/ijerph19084880_

Round 1
Reviewer 1 Report
Major Comments:
- Line 132/133 - so, were all patients with <30 mL/min excluded from the analysis? Or, were they included in category 3 but just not made into a separate group? The authors may kindly clarify this point.
- Does Gender/Sex play a role in the severity of CKD in this cohort? There seems to be a stark increase in percentage from ~46% to ~76% by group 3. Can the authors perhaps comment on this (within the scope of this study)?
- Figures 2 and 3 are basically the same, just with the data arranged differently... The authors can remove this redundancy... Figure 3 could be retained, and the other information from Fig. 2 also be coded into the same.
- Similarly, Fig. 5 is already mentioned in Table 1, and feels redundant.
- Line 222- what are the authors referring to here as overlapping therapy? Is it a combination of Amiodarone + Propafenone? Or is it referring to electrical CV or other treatment perhaps?
- The authors can further discuss why there is an inverse trend between the groups in terms of PCV treated with Antazoline and Propafenone (as in Fig 2) and how this affects treating physician decisions...
Minor Comments:
- Page 1, Line 38 - introduce the abbreviation before usage (CKD)
- Line 171 - ..... older than* patients of .....
- The X-axis on Fig 2b/c are overlapped and not visible - please fix them. It would also be beneficial to add the number of patients in each comparison (perhaps in parentheses?).
- 6 patients were reported for safety endpoint from group III - 4 had bradycardia, 1 had hypotension, what were the causes(s) for the final patient?
Author Response
Thank you very much for a careful analysis of our manuscript, and also for all the notes and the comments. Enclosed please find our responses.
Major Comments:
1.Line 132/133 - so, were all patients with <30 mL/min excluded from the analysis? Or, were they included in category 3 but just not made into a separate group? The authors may kindly clarify this point.
Group III included all patients with eGFR<45 mL/min, also the patients with eGFR<30 mL/min. We have changed the description to be more precise:
"Too little number of patients did not allow us to create a separate group for ones with extremely advanced CKD (<30 mL/min), and therefore all these patients were included into Group III. Otherwise, it might have affected the reliability of the results."
2.Does Gender/Sex play a role in the severity of CKD in this cohort? There seems to be a stark increase in percentage from ~46% to ~76% by group 3. Can the authors perhaps comment on this (within the scope of this study)?
Indeed, there is a noticeable increase in percentage of women in Group III of this cohort. According to your suggestion, we commented on this in the manuscript:
"We found a greater percentage of women in Group III. This phenomenon is difficult to explain since epidemiological data indicate that male patients show a higher incidence of advanced CKD and atrial fibrillation. Presumably, this is specificity of our cohort."
3.Figures 2 and 3 are basically the same, just with the data arranged differently... The authors can remove this redundancy... Figure 3 could be retained, and the other information from Fig. 2 also be coded into the same.
That is true, we fully agree that Figures 2 and 3 present similar data, however such an arrangement allowed us to present the results graphically from two different points of view, and what more important, it allowed us to present a related statistical analysis much more clearly. Therefore, we would appreciate you to accept this form of presentation.
4.Similarly, Fig. 5 is already mentioned in Table 1, and feels redundant.
According to your remark, Figure 5 has been removed.
5.Line 222- what are the authors referring to here as overlapping therapy? Is it a combination of Amiodarone + Propafenone? Or is it referring to electrical CV or other treatment perhaps?
The overlapping therapy means the combination of drugs used in the study (Amiodarone+Propafenone). It does not apply to ECV or any other drugs. We have added an additional explanation in parentheses: "..., or combination of these agents (overlapping therapy)."
6.The authors can further discuss why there is an inverse trend between the groups in terms of PCV treated with Antazoline and Propafenone (as in Fig 2) and how this affects treating physician decisions...
The data presented on Figure 2 show that PCV may be less effective overall in patients with more advanced CKD, although the differences did not reach a statistical significance. The key finding of our study is that antazoline, which appears to be so effective in general population, loses its exceptional abilities in patients with the advanced CKD. However, this does not apply to propafenone. This certainly may affect treating physician decision, and therefore, we have added an additional remarks in the text:
1) "Apparently, it loses its quite unique abilities in CKD environment, and this should be taken into consideration while taking treatment decisions"
2) "Therefore, one may conclude that propafenone, and not antazoline, should be considered first, if we decide to pharmacologically terminate recent-onset AF in patients with advanced CKD."
Minor Comments:
1.Page 1, Line 38 - introduce the abbreviation before usage (CKD)
It has been corrected
2.Line 171 - .....older than* patients of .....
It has been corrected
3.The X-axis on Fig 2b/c are overlapped and not visible - please fix them. It would also be beneficial to add the number of patients in each comparison (perhaps in parentheses?).
The overlapping is presumably the problem of PC software, because in the originally submitted version of manuscript, none of Figures were overlapped in any part. I was informed about such a problem, when I previously submitted a manuscript using Microsoft Word template prepared by IJERPH. I may only suggest to download a PDF version of the manuscript.
We have added the number of patients in each figure caption.
4.6 patients were reported for safety endpoint from group III - 4 had bradycardia, 1 had hypotension, what were the causes(s) for the final patient?
We apologize for this shortcoming - this was a simple typo. Five patients (5.2%) had bradycardia and one (1.0%) had hypotension, which accounts for 6 adverse events in total in the group of eGFR<45 mL/min. We have changed the data in the table and in the text.

Reviewer 2 Report
Ceynowa-Sielawko and co-workers present a pharmacological analysis of the effectiveness of several antiarrhythmic drugs in a clinical setting (a subanalysis of the SCANT II Project published by the same group, Specifically, the analyzed the cardioversion of recent-onset atrial fibrillation (AF) in patients con different severity levels of chronic kidney disease or CKD. The reason is that CKD is present in a significant percentage of AF patients. According to their findings, the first choice treatment for AF antazoline, is not superior to other drugs like propafenone in AF patients with a history of advanced CKD. As far as I know, this seems to be the point made by the authors out of this research. Even if the results are of therapeutic interest for a medical audience, the whole clinical research is not well framed in the manuscript, which makes the reading difficult to follow.
Major concerns:
- Introduction: briefly define cardioversion al the first instance. Please, depict the antiarrhythmic drugs to maintain sinus rhythm according to the Vaughan-Wlliams classification. The authors mentioned the IA and IC drugs and then mentioned amiodarone, which is a III class drug. There is a mess up of different classes of antiarrhythmics. Antazoline is not currently listed in official guidelines to treat paroxysmal AF, please, clarify this point in the introduction. And finally, explain in a few words their mechanism of pharmacological action.
- What is the objective of the study? Please, state it clearly at the end of the introduction.
- Please define the meaning of “paroxysmal” in just two or three words. Please, describe the acronym eCFR. Same for ESC and CHA2DS2.
- There are some edits to be fixed. For instance in line 136: “The primary endpoint was termination….(sic)…”; line 161 “…into a logistic regression model….”; line 211-212 “when the patients of the group III were considered…”; line 213: “…than propafenone…”.
- Methods section: statistics. What was the objective of these particular statistical tests? Please justify their application based on the specific questions to be answered.
- Results section. There are just shown the p values, but there are no results of the conducted analyses. I don't see the simple logistic regression analysis in Table I.
- Table I: It should be written as follows: “Dose (total dose)”. Should not the dose annotation be as mg per kg?
- How do the authors calculate the “% of Success”? SD intervals are not shown in the graphic bars.
- Lines 214-215 the statement that a difference “was not close to being statistically significant (p = 0.067)” is too biased. This difference is NOT significant.
- Discussion. It should be rewritten thoroughly. For instance the first paragraph, which is the most important to draw readers attention, does not match with the title. The discussion section should begin with the sentence written in lines 267-268 and then continue with the pieces of text in lines 256-257 and 284-294. I mean to build a new paragraph of just 6-8 lines explaining the main findings of the research. The second paragraph of the discussion better fits in the Introduction section. The 3rd paragraph of page 9 should go to the Results section and the summary of the pharmacological findings should be included in the first paragraph of the discussion. In sum, the discussion section should be reorganized in depth.
- Figure 2 panels are not well lined up and the vertical axis have slightly different sizes, so comparisons among them are difficult to establish.
Author Response
Thank you very much for a careful analysis of our manuscript, and also for all the notes and the comments. Enclosed please find our responses.
Ceynowa-Sielawko and co-workers present a pharmacological analysis of the effectiveness of several antiarrhythmic drugs in a clinical setting (a subanalysis of the CANT II Project published by the same group). Specifically, the analyzed the cardioversion of recent-onset atrial fibrillation (AF) in patients con different severity levels of chronic kidney disease or CKD. The reason is that CKD is present in a significant percentage of AF patients. According to their findings, the first choice treatment for AF antazoline, is not superior to other drugs like propafenone in AF patients with a history of advanced CKD. As far as I know, this seems to be the point made by the authors out of this research. Even if the results are of therapeutic interest for a medical audience, the whole clinical research is not well framed in the manuscript, which makes the reading difficult to follow.
Major concerns:
1.Introduction: briefly define cardioversion al the first instance. Please, depict the antiarrhythmic drugs to maintain sinus rhythm according to the Vaughan-Wlliams classification. The authors mentioned the IA and IC drugs and then mentioned amiodarone, which is a III class drug. There is a mess up of different classes of antiarrhythmics. Antazoline is not currently listed in official guidelines to treat paroxysmal AF, please, clarify this point in the introduction. And finally, explain in a few words their mechanism of pharmacological action.
We took your comments into account and revised the paragraph:
"Electrical (ECV) or pharmacological (PCV) cardioversion are common methods used world-wide to terminate an atrial fibrillation and to restore a physiological sinus rhythm. Electrical cardioversion requires short-term general anaesthesia and should be preceded by at least 6 hours fasting, thus a pharmacological approach is usually a primary option in most of the cases. Anti-arhythmic drugs (AAD) variously modify ion channel function and/or intracellular mechanisms regulated by adrenergic activity. They are classified according to the Vaughan-Williams classification. Administration of class IA drugs (eg quinidine, disopyramide) or IC (eg propafenone, flecainide), with a dominant sodium channel blocking activity, is often successful, although pro-arrhythmia is a noted phenomenon, particularly in heart failure patients [3]. Amiodarone, class III drug affecting a broad range of channels including potassium, sodium, calcium ones, as well as blocking a- and b-receptors, has a delayed onset of action [4]. New agent vernakalant, with an atrial-selective potassium channel blocking activity, does not actually fit in the Vaughan-Williams classification. It is perceived as a highly effective, but it is not widely available [5]. Antazoline seems to be a reasonable alternative."
2.What is the objective of the study? Please, state it clearly at the end of the introduction.
We have modified the end of the introduction:
"Thus, the objective of this study was to evaluate safety and effectiveness of PCV (primary aim), in particular with intravenous antazoline in comparison to amiodarone, propafenone and/or overlapping therapy (secondary aim), administered in patients with CKD admitted to the emergency department for termination of AF."
3.Please define the meaning of “paroxysmal” in just two or three words. Please, describe the acronym eCFR. Same for ESC and CHA2DS2.
We have defined the meaning of ‘paroxysmal’ ("lasting up to 7 days"), and have described the acronyms eGFR ("estimated glomerular filtration rate"), ESC ("European Society of Cardiology") CHA2DS2-VASc ("scoring system estimating a risk of stroke in patients with AF"), and also TIA ("transient ischemic attack"), BMI ("body mass index"), EHRA ("European Heart Rhythm Association").
4.There are some edits to be fixed. For instance in line 136: “The primary endpoint was termination….(sic)…”; line 161 “…into a logistic regression model….”; line 211-212 “when the patients of the group III were considered…”; line 213: “…than propafenone…”.
All these edits have been fixed.
5.Methods section: statistics. What was the objective of these particular statistical tests? Please justify their application based on the specific questions to be answered.
The Mann-Whitney U test was utilized to compare the statistical significance of inter-group differences in case of non-normally distributed continuous variables, while student's t test applied for normally distributed parameters. Chi-square test was applied for comparing inter-group differences in case of qualitative parameters
6.Results section. There are just shown the p values, but there are no results of the conducted analyses. I don't see the simple logistic regression analysis in Table I.
We have provided the additional Table 2 covering the results of univariate analysis and the results of logistic regression analysis for prediction of successful pharmacological cardioversion in patients with known eGFR.
7.Table I: It should be written as follows: “Dose (total dose)”. Should not the dose annotation be as mg per kg?
We have presented the data concerning the drug doses and the methods of its administration in the Material and Methods section. We emphasized that the allocation of antiarrhythmic drug, its initial and total dose and concomitant medications were left to discretion of attending physician. We based our statistical analysis on a comparison of the total drug dose used to terminate AF, which seems to be the most objective in this case. The initial/partial drug doses varied between physicians and medical centers, and therefore they did not seem to be good/reliable comparative variables.
However, I am taking into account that I understood your comment improperly. If your intention was just to suggest us to correct the phrase we used, then we did it: "Drug (total dose)"
In our clinical management of atrial fibrillation, we usually do not administer drugs in doses calculated per body weight (Zimetbaum P: Antiarrhythmic drug therapy for atrial fibrillation. Circulation 2012, 125: 381-389)
8.How do the authors calculate the “% of Success”? SD intervals are not shown in the graphic bars.
The % of success represents the number of patients who regained sinus rhythm in relation to the total number of patients submitted to specific antiarrhythmic drug. As the success rate represents a qualitative parameter, the value is expressed a crude number of cases of success and percentage. SD should not be used in such instances, but for continuous variables characterized by normal distribution.
9.Lines 214-215 the statement that a difference “was close to being statistically significant (p = 0.067)” is too biased. This difference is NOT significant.
This is true that the difference is not significant. This was only our remark in the Results section that in patients of Group III, antazoline was close to be inferior even to amiodarone. Nevertheless, in the Discussion section we clearly stated that antazoline was non-significantly inferior to amiodarone. After your comment, we have removed this sentence.
10.Discussion. It should be rewritten thoroughly. For instance the first paragraph, which is the most important to draw readers attention, does not match with the title. The discussion section should begin with the sentence written in lines 267-268 and then continue with the pieces of text in lines 256-257 and 284-294. I mean to build a new paragraph of just 6-8 lines explaining the main findings of the research. The second paragraph of the discussion better fits in the Introduction section. The 3rd paragraph of page 9 should go to the Results section and the summary of the pharmacological findings should be included in the first paragraph of the discussion. In sum, the discussion section should be reorganized in depth.
We have reorganized the Discussion section according to your suggestions.
11.Figure 2 panels are not well lined up and the vertical axis have slightly different sizes, so comparisons among them are difficult to establish.
This is presumably the problem of PC software, because in the originally submitted version of manuscript, we exerted an effort the panels to be well lined up and the axes to be the same sizes. I was informed about such a problem, when I previously submitted a manuscript using Microsoft Word template prepared by IJERPH. I may only suggest to download a PDF version of the manuscript.

Reviewer 3 Report
In this paper, authors aimed to investigate a potential role of antazoline for pharmacological cardioversion in AF e CKD patients, compared to propafenone and amiodarone. The drug choicen is not widespread used in the world, but I believe that it could be interesting for those countries where it is used. Some comments:
-
patients with CKD stage III with GFR < 45 are older and with more comorbidities, this could explain the less efficacy of antazoline more than the CKD alone. Is there a reduced dosage of antazoline for CKD patients? Please specify.
-
moreover, I will suggest to discuss not only the CKD stages in relation to the results found, but also the concept that having patients with CKD in advanced stages is consistent with the possibility of having older patients, more hypertension and so far...so with more comorbidites that could play a role in the les efficacy of the drug.
-
In the Introduction, it will be useful to discuss the importance of AF in the contest of primary prevention, as well expressed here: "Mazzone C, Prognostic role of cardiac calcifications in primary prevention: A powerful marker of adverse outcome highly dependent on underlying cardiac rhythm. Int J Cardiol. 2018 May 1;258:262-268", where in subjects with multiple CV risk factors the presence of AF nullifies the prognostic power of cardiac calcifications, conversely to patients in sinus rhythm.
-
an important predictor of success for AF cardioversion is the BMI of patients, as expressed here "Lip GYH, Clinical factors related to successful or unsuccessful cardioversion in the EdoxabaN versus warfarin in subjectS UndeRgoing cardiovErsion of Atrial Fibrillation (ENSURE-AF) randomized trial. J Arrhythm. 2020 Apr 15;36(3):430-438", however authors didn't show this data in their analysis. Could you add this? Or if not available, could you discuss it as a main limitation?
Author Response
Thank you very much for a careful analysis of our manuscript, and also for all the notes and the comments. Enclosed please find our responses.
In this paper, authors aimed to investigate a potential role of antazoline for pharmacological cardioversion in AF e CKD patients, compared to propafenone and amiodarone. The drug choicen is not widespread used in the world, but I believe that it could be interesting for those countries where it is used. Some comments:
1.Patients with CKD stage III with GFR < 45 are older and with more comorbidities, this could explain the less efficacy of antazoline more than the CKD alone. Is there a reduced dosage of antazoline for CKD patients? Please specify.
No, there is not. According to Summary of Product Characteristics, there is no need to reduce antazoline dose in patients with CKD, when it is administered intravenously to terminate atrial fibrillation.
2.Moreover, I will suggest to discuss not only the CKD stages in relation to the results found, but also the concept that having patients with CKD in advanced stages is consistent with the possibility of having older patients, more hypertension and so far...so with more comorbidites that could play a role in the les efficacy of the drug.
We fully agree that older age and comorbidities could play a role in lesser efficacy of antazoline. We already mentioned this in abstract, results and study limitations. According to your suggestion, we discussed it also in the Discussion section:
"This was rather not surprising that patients with the most advanced CKD were older and had more comorbidities; all these elements are biologically and medically closely related to each other. And indeed, this could have had an impact on the treatment results. It is worth noting, however, that antazoline was the only drug with a worsening efficacy profile. Thus, one may not exclude that aging and comorbidities affect the efficiency of antazoline more than they do on other AAD."
3.In the Introduction, it will be useful to discuss the importance of AF in the contest of primary prevention, as well expressed here: "Mazzone C, Prognostic role of cardiac calcifications in primary prevention: A powerful marker of adverse outcome highly dependent on underlying cardiac rhythm. Int J Cardiol. 2018 May 1;258:262-268", where in subjects with multiple CV risk factors the presence of AF nullifies the prognostic power of cardiac calcifications, conversely to patients in sinus rhythm.
According to your suggestion, we have added relevant information in the Introduction section (with compatible references):
"It is influential enough to reduce the prognostic power of cardiovascular event predictors, but above all, it significantly increases the risk of all-cause death."
4.An important predictor of success for AF cardioversion is the BMI of patients, as expressed here "Lip GYH, Clinical factors related to successful or unsuccessful cardioversion in the EdoxabaN versus warfarin in subjects UndeRgoing cardiovErsion of Atrial Fibrillation (ENSURE-AF) randomized trial. J Arrhythm. 2020 Apr 15;36(3):430-438", however authors didn't show this data in their analysis. Could you add this? Or if not available, could you discuss it as a main limitation?
Yes, there are quite a few articles showing that body weight or BMI are important predictors of atrial fibrillation as well as successful cardioversion. The data on BMI of our patients are available. They were analyzed and were presented in Table 1 (no significant differences were found between the studied groups of patients with atrial fibrillation) and now, the additional analyses were presented in a new Table 2 (BMI was not a predictor of successful cardioversion).

Reviewer 4 Report
I would like to thank the authors for their work. The purpose of this article is clearly stated. I would like to add minor comments, which are itemized below.
- Introduction: The authors should summarize the initial hypothesis under study, primary aim and secondary aims.
- Discussion should initially provide an overview of the study and then compare the findings in relation to previous studies.
Good luck!
Author Response
Thank you very much for a careful analysis of our manuscript, and also for all the notes and the comments. Enclosed please find our responses.
I would like to thank the authors for their work. The purpose of this article is clearly stated. I would like to add minor comments, which are itemized below.
1.Introduction: The authors should summarize the initial hypothesis under study, primary aim and secondary aims.
We have modified the end of the introduction clearly indicating what is the objective of this study:
"Thus, the objective of this study was to evaluate safety and effectiveness of PCV (primary aim), in particular with intravenous antazoline in comparison to amiodarone, propafenone and/or overlapping therapy (secondary aim), administered in patients with CKD admitted to the emergency department for termination of AF."
2.Discussion should initially provide an overview of the study and then compare the findings in relation to previous studies.
We have reorganized the Discussion section according to your suggestions.

Round 2
Reviewer 1 Report
The revised version is quite improved, with the introduction section also reading better now. They have addressed most of my comments sufficiently.
Author Response
Thank you again very much for taking the time to review our manuscript; we appreciate all your comments and remarks.
Reviewer 2 Report
The authors have adequately addressed all my questions. The manuscript has been improved through the review process. it basically sounds.
Just two minor stylistic issues:
Line 192: "This is a particular..." or "This is specific to.."
Line 334. "Antazoline proved to be..." (past tense)
Author Response
The authors have adequately addressed all my questions. The manuscript has been improved through the review process. it basically sounds.
Just two minor stylistic issues:
Line 192: "This is a particular..." or "This is specific to.."
Line 334. "Antazoline proved to be..." (past tense)
Thank you very much for taking the time to review our manuscript again; we appreciate all your comments and remarks. It seems that the lines of our version of the manuscript do not match yours (presumable PC software problems), however we have found these stylistic issues and we have corrected them.
Reviewer 3 Report
Authors have already well addressed my previous comments. Just a remarkable note on the appropriate reference suggested that was not reported about their sentence: "It is influential enough to reduce the
prognostic power of cardiovascular event predictors, but above all, it significantly increases the risk of death [2,3]", whereas I had also suggested: "Mazzone C, Prognostic role of cardiac calcifications in primary prevention: A powerful marker of adverse outcome highly dependent on underlying cardiac rhythm. Int J Cardiol. 2018 May 1;258:262-268", where in subjects with multiple CV risk factors the presence of AF nullifies the prognostic power of cardiac calcifications, conversely to patients in sinus rhythm.
Author Response
Authors have already well addressed my previous comments. Just a remarkable note on the appropriate reference suggested that was not reported about their sentence: "It is influential enough to reduce the prognostic power of cardiovascular event predictors, but above all, it significantly increases the risk of death [2,3]", whereas I had also suggested: "Mazzone C, Prognostic role of cardiac calcifications in primary prevention: A powerful marker of adverse outcome highly dependent on underlying cardiac rhythm. Int J Cardiol. 2018 May 1;258:262-268", where in subjects with multiple CV risk factors the presence of AF nullifies the prognostic power of cardiac calcifications, conversely to patients in sinus rhythm.
Thank you very much for taking the time to review our manuscript again; we appreciate all your comments and remarks. We apologize, we made a mistake and we placed improper reference. Thank you for the note; we have corrected this shortcoming.